Potential environmental risk assessment of di-2-ethylhexyl phthalate emissions from a municipal solid waste landfill leachate

Wowkonowicz Paweł pawel.wowkonowicz@ios.edu.pl 1
Kijeńska Marta 1
Koda Eugeniusz 2
1 Environmental Chemistry and Risk Assessment Department, Institute of Environmental Protection - National Research Institute, IOS-PIB , Warsaw , Poland
2 Institute of Civil Engineering, Warsaw University of Life Sciences, SGGW , Warsaw , Poland
Anderson Todd
Electronic publication date: 2021 Oct 1
Publication date: 2021
Volume: 9
Electronic Location ID: e12163
Received 2021 Apr 7; Accepted 2021 Aug 25
Copyright: ©2021 Wowkonowicz et al.
Copyright year: 2021
Copyright holder: Wowkonowicz et al.
License: This is an open access article distributed under the terms of the Creative Commons Attribution License, which permits unrestricted use, distribution, reproduction and adaptation in any medium and for any purpose provided that it is properly attributed. For attribution, the original author(s), title, publication source (PeerJ) and either DOI or URL of the article must be cited.
License URL: https://creativecommons.org/licenses/by/4.0/

Keywords: DEHP, Leachate, Landfill, Risk assessment, EUSES, Phthalate, Phthalic acid esters, PAE, di-2-ethylhexyl phthalate, Risk

Funding: The Polish Ministry of Science and Higher Education 10-OZ-OZ-1455/14 10-OZ-BI-1492/15 10-OZ-BI-1516/16 The work presented was supported by the Polish Ministry of Science and Higher Education, grants numbers 10-OZ-OZ-1455/14, 10-OZ-BI-1492/15 and 10-OZ-BI-1516/16. The funders had no role in study design, data collection and analysis, decision to publish, or preparation of the manuscript.

==============================
Background

In certain countries, including Poland, polyvinyl chloride (PVC) waste, together with di-2-ethylhexyl phthalate (DEHP) contained within (up to 60%), is mostly directed to municipal solid waste (MSW) landfills. From there, over time, it is released from the polymer matrix and can migrate with landfill leachate into the environment. The amount of DEHP placed on the Polish market since the start of industrial production and the prevalent landfilling disposal of PVC waste in Poland, indicate that DEHP pollution can increase risk factors in the future. The objective of this study was to determine the concentrations of DEHP in leachates from a chosen MSW landfill directed to a local sewage treatment plant (STP) and estimate the associated potential risks to the environment.

Results

DEHP concentrations in leachates ranged from < LOQ to 394.4 µg/L, depending on the sampling year and season. DEHP is a pervasive environmental contaminant present in all investigated landfill leachate samples. The results from The European Union System for the Evaluation of Substances (EUSES) modelling related to DEHP in leachate directed to STP indicated potentially unacceptable risk to freshwater organisms; and birds and mammals feeding on earthworms (where a sewage sludge applications in agriculture take place). The results indicated low risk for other environmental components including local fresh-water sediment, local soil and microorganisms of STP, and freshwater fish-eating birds and mammals.

Conclusions

Future DEHP emissions may occur after the technical lifetime of the landfill and/or decay its bottom sealing. To avoid contamination, the monitoring of landfills after closure should include DEHP concentrations and last longer than the recommended (inter alia in Poland) 30 years, or until emissions from PVC to leachate are eliminated. More research on leachate of DEHP and its potential risks should be conducted, utilising detailed modelling which can including other landfills and different routes of DEHP emissions in leachates.

Introduction

The most popular and widely used phthalic acid esters (PAE) in recent years has been DEHP, accounting for one third of the phthalates produced in the EU and 80% produced in China (Gao et al., 2016a; Gao et al., 2016b; Meng et al., 2014). Up to 95% of phthalate production is used in PVC compounds (Mersiowsky, Weller & Ejlertsson, 2001). The percentage of PAE in PVC can reach as much as 60% (Chao & Cheng, 2007; Erythropel et al., 2014). PAEs are attached to the polymer matrix only through physical bonding, and are therefore able to migrate to the surface of the product where these are released into the surrounding environment (Net et al., 2015a; Sharma & Kaur, 2020).

PAEs are currently the most common chemical products that people come into contact (Net et al., 2015a) and have been detected in all environmental compartments, including the air (Li et al., 2017; Wensing, Uhde & Salthammer, 2005), soil, sewage sludge, leachate from landfills (Horn et al., 2004; Kalmykova et al., 2014; Schwarzbauer et al., 2002; Zheng et al., 2009; Zheng et al., 2007; Zolfaghari et al., 2014), and ground and surface water (Schwarzbauer et al., 2002). Among all PAEs, DEHP is considered the most common (Gao & Wen, 2016). DEHP was detected in 100% of sludge samples at concentrations of 11.2 to 275 µg/kg (dry weight) (Gao et al., 2016a; Gao et al., 2016b). In a recent study on wastewater in Poland by Kotowska, Kapelewska & Sawczuk (2020), DEHP was detected in 97% of all influent wastewater samples with the highest concentrations up to 143 mg/L. According to Bauer & Herrmann (1997), the main sources of DEHP as a soil contaminant are the disposal of industrial and municipal waste to landfills and sewage sludge application on soils.

DEHP is classified as an endocrine disruptor (Nagorka & Koschorreck, 2020; Velazquez et al., 2019; Lind & Lind, 2018). The greatest concern in human and animal exposure are carcinogenicity and potential negative impacts on reproduction, including problems with fertility and juvenile development (Howdeshell, Rider & Gray, 2008; Katsikantami et al., 2016; Net et al., 2015b). PAEs are suspected of interfering with biological processes in humans and animals, potentially being teratogenic, mutagenic and carcinogenic (Sharma & Kaur, 2020), even at very low concentrations (Becker et al., 2004; Caldwell, 2012).

To protect humans from exposure to DEHP, various environmental agencies have the authority to regulate its concentrations in drinking water. The United States Environmental Protection Agency (USEPA) has limited their amount to 6 g/l (USEPA, 2020) and the World Health Organization (WHO) has set their maximum safe concentration to 8 g/l (WHO, 2017). DEHP in accordance with Polish and European legislation has been classified as a priority substance for which limits have been set to a maximum of 1.3 µg/L in surface waters (Rozporządzenie Ministra Środowiska, 2016; European Commission, 2013). In addition, EU and Danish legislation have set a safe limit for DEHP content in sewage sludge at levels of 100 and 50 mg/kg dry weight, respectively (Inglezakis et al., 2014).

DEHP is lipophilic (log Kow ∼ 7.5) and will strongly adsorb to organic matter, soil or rock media (Lee et al., 2020), thus accumulating in municipal landfills (Liu et al., 2010; Zolfaghari et al., 2014). Even in adsorbed substances, DEHP may leave the landfill via particle transport in leachates (De Bruijn et al., 2003).

Previous studies on phthalate concentrations in leachate from municipal landfills have been conducted in Europe and worldwide. A summary of those studies is presented in Table 1.

Table 1 Occurrence of phthalates in MSW landfill leachate based on the data presented in literature (µg/L).

Sampling site	Sampling date (comments)	DEHP (µg/L)	Reference	
3 landfills, Podlasie, Poland	2020	n.d.–249	Kotowska, Kapelewska & Sawczuk (2020)	
1 landfill - Shanghai, China	2017 (7 years old landfill)	260.9	Fang et al. (2018)	
1 landfill –Zhejiang, China	2015 (5 years old landfill)	0.46	He et al. (2015)	
4 landfills –Gothenburg region, Sweden	2013	n.d–23	Kalmykova et al. (2014)	
1 landfill - Anasco, Puerto Rico	2012	n.d.–285	Zolfaghari et al. (2014)	
1 landfill –Thailand	2012	65.5	Boonyaroj et al. (2012)	
2 landfills in Shanghai, China	04-07/2007	40–46	Zheng et al. (2009)	
1 landfills in Wuhan, China	12/2007	n.d.–7.2	Liu et al. (2010)	
1 landfill -Montreal, Canada	2004	62	Horn et al. (2004)	
2 landfills in Japan	2000–2001	9.6–49	Asakura, Matsuto & Tanaka (2004)	
11 landfills in Finland	1998–2001	1–89	Marttinen, Kettunen & Rintala (2003a) and Marttinen et al. (2003b)	
3 landfills in Göteborg, Sweden	–	97–346	Zolfaghari et al. (2014)	
Landfills - Bavaria, Germany	–	26.4–240	Zolfaghari et al. (2014)	
7 landfills/landfill cells, Sweden	08/1998	<1–9	Jonsson et al. (2003)	
6 landfills, Denmark	02/1999	<1–3	Jonsson et al. (2003)	
2 landfills, northern Germany	02/1998	<1–≤ 20	Jonsson et al. (2003)	
2 landfills, Italy	09/1998	88–460	Jonsson et al. (2003)	
Notes.

n.d. not detected or < LOQ

– no data available

A landfill site emits leachate throughout its lifetime (increasing emissions with age) and for several hundred years after its closure (Zheng et al., 2007; Wang, Pelkonen & Kaila, 2012). Although it is assumed that the amount of DEHP released from landfills today is small, future emissions from municipal landfills may increase (Pakalin et al., 2008). The technical guarantee for landfill leachate collection systems (bottom liners and pipes) is restricted to 80 years (Spillmann, 2000). However, it is also expected that DEHP will be released from waste for many years (Asakura, Matsuto & Tanaka, 2004) and DEHP emissions are likely to last longer than the technical barrier (De Bruijn, 2003; Koda & Osinski, 2017), posing a threat to the environment (Mishra et al., 2016; Sieczka et al., 2019).

With an average lifetime of around 30 or more years for PVC products (EU Commission, 2000), the quantity of PVC in wastes is still considered very small compared to PVC consumption. Large quantities of PVC waste were expected to appear around 2010 and will increase drastically after 2020 (Plinke et al., 2000). Reuse or recovery operations involving soft PVC are practically impossible (Lassen et al., 2009) and the thermal disposal of PVC is also subject to restrictions (Wasielewski & Siudyga, 2013). It is assumed that in many countries, including Poland, PVC waste together with DEHP contained has been directed to the MSW landfills. More than 1 million Mg of DEHP may have been placed on the Polish market since the start of the mass DEHP production in 1986. It is expected that DEHP pollution will pose a problem in the future given its pervasive disposal in landfills.

If landfills have proper sealing and leachate collection systems, leachate should be transported to municipal sewage treatment plants. In this case, wastewater and sewage sludge have been identified as the main ways of introducing DEHP into the environment (Zolfaghari et al., 2014).

However, current Polish environmental law states no legal obligation to test landfill leachate, sewage sludge, or wastewater for DEHP content.

The authors argue that DEHP concentrations in landfill leachate pose a threat to the environment, even in the case of a controlled landfill, where all leachate (considered as a single source of DEHP) is directed to STPs.

In this study environmental risk assessment (ERA) was conducted with the use of EUSES 2.1.1 (The European Union System for the Evaluation of Substances), which was specifically developed for quantitative risks assessment of new and existing chemical substances and biocides to humans and the environment. EUSES is the recommended risk assessment modelling tool by the European Technical Guidance Document (TGD) (De Bruijn et al., 2003) and can specifically be used in the initial screening and intermediate stages of assessments. Initial screenings can determine if more data are required and if a more refined assessment is necessary (Ladefoged, Nielsen & Muller, 2004). The risk analysis was based on the calculation of risk characterisation ratios (RCRs) by comparing the PECs (predicted environmental concentration) to the PNECs (predicted environmental no-effect concentration).

In summary, the objective of this study was to determine the DEHP emissions from one MSW landfill in Pruszków over a 3-year period, investigate the seasonal changes of DEHP concentrations in its leachate, and estimate, with the use of EUSES modeling, the associated risks to environmental components.

Materials & Methods

Di (2-ethylhexyl) phthalate (DEHP), CAS number 117-81-7 was chosen in this study because of its toxicity, historical popularity in the environment.

Characteristics of sampling sites and location

A quarter of the municipal solid waste (MSW) landfill “MZO Pruszków” was chosen for this research. Permission to obtain samples was granted by the landfill management on 11.12.2013. The landfill has been used for the disposal of non-hazardous and inert waste from neighbouring towns and the capitol city Warsaw, but there is no precise record of its contents. The landfill opened in 1965 and is located in Mazowieckie Voivodship, near Pruszków city, 20 km from the center of Warsaw, the capital of Poland (52°10′40.0″N 20°46′34.4″E) (Fig. 1). The investigated quarter of the landfill was opened in 2007, it has synthetic bottom isolation, and systems for controlling and collecting leachate in a temporary 48 m3 roofed storage tank. The 35 m high quarter, which is currently under a closing process, occupies around 1 ha and contains nearly 447000 m3 of waste. The leachate is transported to a local STP located 2 km away from the landfill.

Figure 1 Location of landfill site.

Contour maps: Wikimedia Commons, Creative Commons Attribution-Share Alike 3.0 Unported license. Photos credit: Paweł Wowkonowicz.

Sampling and quality control/quality assurance

Leachate samples were collected three times per year over three years from 2014 to 2016 from the landfill. Sampling took place in each of the four “meteorological seasons” to examine seasonal variation on ambient temperature, precipitation, evaporation and water content in the waste, which could affect DEHP concentrations.

Irish EPA guidelines (Cambell et al., 2003) were used during the collection of three separate samples of raw leachate. Whenever possible, individual samples were taken at different locations and depths across and within the raw leachate tanks.

In 2015 and 2016 two piezometers installed by the landfill managers upstream and downstream of an aquifer near the landfill for monitoring purposes were used to collect groundwater samples. Water samples from the piezometers were collected using a submersible pump with a maximum capacity of 9 L/min. Water was collected after cleaning the piezometers by pumping out stagnant water in the borehole several times or after pumping out water from the borehole for a specified time. Guidelines for groundwater sampling in accordance with PN-EN ISO 5667-11:2004 were followed.

All samples were collected using a stainless steel bucket, Teflon tubes and clean glass jars. A portion test material was poured to the top of the jars to eliminate oxygen contact during transportation. After collection, the samples were immediately sent to the laboratory for chemical analysis. No plastic materials were used during sampling, transportation or analysis to avoid contamination. Glass jars and aluminium foil (used for insulation under the jar’s cap) were first washed with water and detergent solution then rinsed several times with distilled and Milli-Q water and dried for several hours in a 200 °C oven. All jars were also rinsed with water or leachate before sample collection.

Overall there were 26 leachate and 19 water samples collected and analysed in the study. The collected leachates were yellowish brown to dark brown in color. Table 2 presents weather details and seasons of sampling.

Table 2 Details of the sampling times and weather condition.

Year and series	Season of sampling	Weather conditions	
2014 - I	Summer	22 °C, no precipitation, sunny and dry	
2014 - II	Autumn	5 °C, no precipitation, cloudy and dry	
2014 - III	Winter	3 °C, light rain, last precipitation occurred during the night before samplings	
2015 - I	Summer	25 °C, no precipitation, sunny and dry	
2015 - III	Autumn	3 °C, no precipitation, cloudy and dry	
2015 - III	Winter	2 °C, no precipitation, cloudy, last precipitation occurred 2 days before samplings	
2016 - I	Spring	10 °C, no precipitation, sunny, precipitation occurred 3 days before samplings	
2016 -II	Summer	18 °C, no precipitation, sunny, last precipitation occurred 2 days before the sampling (heat wave of 34 °C before samplings)	
2016 -III	Autumn	7 °C, no precipitation, sunny, last precipitation occurred 2 days before the sampling, large precipitation noted 2 weeks before samplings	

Instrumental analysis

The first three series of leachate samples in 2014 (summer, autumn and winter) were analysed using GC–ECD methodology and all the subsequent series in 2015 and 2016 were analysed using GC-MS methodology described below. The main reason for the change was that GC-MS methodology was accredited and it was acknowledged that international accreditation would ensure the highest quality of analytical results. Details about each methodology can be found in the references cited and in the supplementary materials (Supplemental Data S1).

GC–ECD methodology

The leachate samples were placed in 1 litre glass separators. 100 mL of ethyl acetate were then added and extracted by a liquid-liquid method. The extract was dried on glass funnels filled with anhydrous sodium sulphate and concentrated in a stream of nitrogen to about 30 mL. one mL of the solution was taken to a 1.5 mL vial and analysed for DEHP content using the GC–ECD methodology, which employed Varian liquid chromatographs with ECD detection. Phthalates were separated on VF-Xms (Arylene/methyl modified polysiloxane) 30 m × 0.25 mm ID × 0.39 mm, 0.25 m chromatographic column. The temperature program of the furnace was 70 °C (for 3 min isothermally) to 280 °C with an increase of 13 °C per minute for 20 min. Helium was used as a carrier gas with a constant flow of one mL/min. The temperature of the dispenser was 250 °C, the injection was 1L, and the detector temperature was 300 °C. A qualitative analysis of phthalates was based on retention times, and a quantitative analysis was based on signals (peaks) using the calibration curve method. Limit of quantification (LOQ) of the method was established at the level of 1 g/L. The method was validated with respect to linearity, precision and accuracy. The recoveries ranged from 50% to 100%. The precision of the method ranged from 15% to 30%. Precision and recoveries were made with each measurement series.

GC–MS methodology

The accredited method (The Certificates of Accreditation no. 819/2015 and 319/2016) was based on US EPA 8061A (USEPA, 1996), 3500 (USEPA, 2007a), 3510 (USEPA, 2007b) methods. The samples were analysed as previously described in Wowkonowicz & Kijeńska (2017). Specifically, to each 500 mL of sample, an extraction standard FTA-ISTD10 of acetone was added. The sample was then extracted twice with 30 mL dichloromethane. The extract was then transferred into hexane and concentrated to 0.25 mL. 1 L of the prepared solution was analysed.

Gas chromatography coupled with mass spectrometry (model Agilent 7890/5975C) was used for the analysis. The sample was injected in splitless mode at 250 °C. The separation was performed on the column at DB-5MS: 20 m (length), 0.18 mm (diameter), 0.18 m (thickness of a phase in the column), and ion m/z 149. Five point calibration was used in the range of 0.5–10 g/mL.

The following standards were used for the calibration:

ISTD: 10 mg/mL, custom mix of deuterate phthalates, Chiron S-4727-10K-AC

LCS: 2/20 mg/mL, phthalate standard 12 components, Absolute standards, 97625

Calibration: 2/20 mg/mL, organic standard solution, Chromservis, 3389.20 K. A. 1.5 syringe std. neat.

For all measured values uncertainty was +/- 35% and LOQ was 1.3 g/L. For some of the samples the LOQ was raised. Landfill leachate is rich in dissolved organic matter, inorganic compounds, heavy metals and xenobiotic organic materials, thus the complexity of DEHP analyses and the multitude of potential difficulties (such as matrix interference) likely account for these results.

EUSES modelling

During our modelling using EUSES program default data from the defined standard environment, main DEHP characteristics such as physicochemical properties (source: Pakalin et al., 2008) and local specific data (obtained DEHP concentrations and emissions, local measured temperatures and precipitations) were used to calculate the predicted PEC environmental concentrations on a local scale.

For a multitude of environmental compartments including freshwater bodies and sediments, STP sludge, soil, and for secondary poisoning, the endpoints and PNEC values have been developed at the European level and published in the ECHA report (ECHA database, 2021) and the DEHP European Union Risk Assessment Report (EU RAR) (Pakalin et al., 2008). According to the report no long term studies indicating effects on aquatic organisms exposed to DEHP existed at the time of EU RAR (Pakalin et al., 2008), therefore PNECwater  could not be specified. However, the results of a 2013 long-term study of the apparent water solubility of DEHP were deemed acceptable, in which DEHP concentrations of 0.2 g/L impaired reproduction in zebra fish (Danio rerio): thus the PNECwater was determined to be 0.07 g/L (Corradetti et al., 2013). This test, as well as the PNECwater, were also found to be reliable and were therefore used in the PAE risk assessment performed by the Government of Canada in 2017 (Environment and Climate Change Canada, 2017). All collected endpoint and PNEC values are summarized and presented in Table 3.

Table 3 End points and PNEC.

Environmental compartment	End points	PNEC	Toxicity toward	Sources	
Freshwater	CTV*= 0.0002 mg/L	PNECfreshwater	0.00007 mg/L	fish: Danio rerio	Corradetti et al. (2013); Health Canada (2020)	
Sediment (freshwater)	NOEC = 1000 mg/kg dw.	PNECSediment (freshwater)	100 mg/kg sediment dw.	frog eggs	ECHA database (2021)	
STP microorganisms	NOEC = 2007 mg/L	PNECSTP	201 mg/L	microorganisms		
Soil	NOEC = 300 mg/kg soil dw	PNECsoil	13 mg/kg soil dw	soil microorganisms	ECHA database (2021)	
Secondary poisoning	NOEC = 160 mg/kg food	PNECoral, fish	16 mg/kg food	fish	Wood and Bitman (1980); Pakalin et al. (2008)	
Secondary poisoning	NOEC = 1700 mg/kg food	PNECoral, birds	3.3 mg/kg food	birds	ECHA database (2021)	
Secondary poisoning	NOEC = 33.3 mg/kg food	PNECoral, mammals	3.3 mg/kg food	rats	Wolfe et al. (2003)	
Notes.

CTV critical toxicity value

The outcomes of EUSES modelling were PEC and RQ values.

If RQ (PEC/PNEC) < 1, the substance is not considered of concern and no unacceptable risk to the environment is identified. However, if the RQ ratio > 1, a potentially unacceptable risk is identified and further testing and risk reduction measures should be considered (Gruszecka & Helios-Rybicka, 2009).

Results

DEHP concentrations in groundwater and landfill leachate

In groundwater, in 2015 and 2016 all samples upstream and downstream piezometer’s water resulted in DEHP concentrations below LOQ, except for one sample in autumn 2016 of 1.5 g/L (Table 4).

Table 4 DEHP concentrations in groundwater, raw MSW landfill leachate and precipitation in the region.

Year	Season of sampling	DEHP concentration in groundwater (µg/L)	DEHP concentration in leachate (µg/L)	Mean DEHP concentration in leachate (WCS) (µg/L)	Average precipitation per season** (mm)	Total annual precipitation** (mm)	
		upstream	downstream					
2014	Summer	–	–	18.5	200,4	81	622,4	
		–	–	22.3				
		–	–	17.8				
	Autumn	–	–	256.7		16		
		–	–	394.4				
		–	–	–				
	Winter	–	–	167.0		44		
		–	–	184.3				
		–	–	22.5				
2015	Summer	<LOQ	<LOQ	64.9*	61,5	33	407,6	
		<LOQ	<LOQ	65.4				
		<LOQ	<LOQ	73.9*				
	Autumn	<LOQ	–	30.2		45		
		<LOQ	–	58.1*				
		<LOQ	–	9.1*				
	Winter	<LOQ	<LOQ	52.6*		36		
		<LOQ	<LOQ	44.6				
		<LOQ	<LOQ	34.4*				
2016	Spring	–	–	43.1*	23,3	30	667,6	
		–	–	36.0				
		–	–	32.4*				
	Summer	–	–	7.5		75		
		–	–	12.0				
		–	–	7.1				
	Autumn	1.5	–	<LOQ		63		
		<LOQ	–	<LOQ				
		<LOQ	–	<LOQ				
Notes.

WCS a worst-case scenario

– no data available

* source: Wowkonowicz & Kijeńska, 2017.

** based on the data from meteorological station provided by the Institute of Meteorology and Water Management - National Research Institute (IMGW-PIB).

The results of DEHP concentrations in raw leachate are presented in Table 4 and Fig. 2. DEHP was detected in 88.5% of leachate samples from the landfill (24 out of 27 cases).

The average concentration in each year (Table 4) was calculated using highest observed DEHP concentrations from each sampling season while assuming a worst-case scenario (WCS) for further calculation. The highest average concentration was in 2014 at 200.4 µg/L; 2015′s highest average concentration was 61.5 µg/L; and 2016 had the lowest value at 23.3 µg/L.

Seasonal changes

Average seasonal precipitation was calculated based on data from the nearest weather station in Pruszków. From these data annual precipitation were calculated (for more details please refer to Table 4). The highest sums of annual precipitation of 667.6 mm were observed in 2016; the lowest in 2015 of 407.6 mm; while 2014 held a value of 622.4 mm.

Risk assessment

For the purpose of modelling, the predicted main routes of DEHP emissions in leachates from Mazowieckie Voivodeship municipal landfills to the environment and an estimated percentage of total DEHP amounts were developed by the authors and are presented in Fig. 3. The storage of sewage sludge on the STP premises is assumed to be a temporary condition, WCS was assumed in the risk analysis of sewage sludge agricultural applications. Considering the information presented in Fig. 3, it may be assumed that from municipal landfills without bottom sealing and a leachate collection system, 100% of the DEHP load in leachate will make its way to the environment. In contrast, for municipal landfills with bottom sealing and leachate collection, 6.8% of the DEHP phthalate load could be released directly to the environment from municipal wastewater treatment plants; 11% of the load is dumped into the environment as a result of sewage sludge storage and 4.6% is transferred to the environment as a result of legal agricultural use of sewage sludge. Based on our calculations 22.4% of the total DEHP load was destined for additional analyses and EUSES modeling.

Figure 2 DEHP concentrations in raw MSW landfill leachate (µg/L).

Figure 3 Routes of DEHP leachate emission from municipal landfills in the Mazowieckie Voivodeship to the environment, together with estimated loads.

The authors’ calculations based on available data (De Bruijn et al., 2003; Gadomska et al., 2018).

Possible routes of DEHP emissions in leachate from municipal landfills (with and without bottom sealing) into the environment were analysed and a risk assessment algorithm was developed by the authors (Fig. 4).

Figure 4 Risk analysis algorithm for DEHP leachate emissions from municipal landfills to different elements of the environment.

Algorithm developed by the authors.

Using EUSES modelling PEC values (for the sewage sludge application of 3 and 15 Mg/ha/year) and RCR values were calculated and are presented in Tables 5 and 6 respectively. RCR >1 for the local fresh-water compartment (RCR of 16.1) and for worm-eating birds and mammals (RCR of 8.63) for a maximum sewage sludge application in agriculture of 3 Mg/ha/year. RCR of 43.2 was obtained for worm-eating birds and mammals for a maximum sewage sludge application for other purposes, such as: reclamation of land for non-agricultural purposes; cultivation of plants intended for compost production; and cultivation of plants not intended for consumption or production of fodder (15 Mg/ha/year).

The results of this study indicated RCR values below 1 for compartments including local fresh-water sediment, local soil and microorganisms of a sewage treatment plant, and fish-eating birds and mammals (fresh-water).

For more information please refer to the full EUSES Reports included in Supplemental Data S2 and Supplemental Data S3.

Discussion

DEHP concentrations in groundwater and landfill leachate

Our results suggests no DEHP contamination of groundwater, which means the landfill’s synthetic bottom isolation, and systems for controlling and collecting leachate, continues to work properly.

The most recent study on leachates from active and closed landfills in Poland (Kotowska, Kapelewska & Sawczuk, 2020), also reported similar DEHP concentrations in the range of LOQ - 249 g/L and LOQ - 143 g/L respectively, but other Polish researchers (Fudala-Ksiazek, Pierpaoli & Luczkiewicz, 2017) reported much higher DEHP concentrations: 8201 g/L. The variability of these results suggests that DEHP concentrations in MSW landfills’ leachate can be large. Many reasons may account for such difference, such as local precipitation, ambient temperature and water content in the waste, sampling season, landfill age and design (including the methods of compacting, top cover and type of surface vegetation), capture and storage of leachate (open or close tanks) and the related surface runoff and dilution of leachate; therefore concentrations may differ across countries as well as individual landfills (Kotowska, Kapelewska & Sawczuk, 2020; Marttinen, Kettunen & Rintala, 2003a; Renou et al., 2008). The range of raw leachate DEHP concentrations in this study are similar to those found in the scientific literature (Table 1), where concentrations ranged from “not detected” to 460 g/L. The level of DEHP in the leachate in our study was similar to those reported in China (Feng et al., 2018), Puerto Rico, Germany, Sweden (Zolfaghari et al., 2014) and Canada (Horn et al., 2004). Our analyses showed DEHP concentrations much higher than those reported on two landfills in China (Liu et al., 2010; He et al., 2015), eleven landfills in Sweden (Kalmykova et al., 2014; Jonsson et al., 2003) and six landfills in Denmark (Jonsson et al., 2003). The difference in DEHP concentrations could be also attributed to the composition of landfilled waste, mainly the amounts and types of plasticizers used in Asia, Western and Eastern Europe.

Table 5 Calculated PEC values.

	PEC value	Units	
Environmental compartment			
Local PEC in surface water during emission episode (dissolved)	1.12	(µg/L)	
Local PEC in fresh-water sediment during emission episode	4.03	(mg/kgwwt)	
Sewage sludge application of 3 Mg/ha/year	
Local PEC in agric. soil (total) averaged over 30 days	1.57	(mg/kgwwt)	
Local PEC in agric. soil (total) averaged over 180 days	1.55	(mg/kgwwt)	
Local PEC in groundwater under agricultural soil	0.53	(µg/L)	
Sewage sludge application of 15 Mg/ha/year	
Local PEC in agric. soil (total) averaged over 30 days	7.83	(mg/kgwwt)	
Local PEC in agric. soil (total) averaged over 180 days	7.74	(mg/kgwwt)	
Local PEC in groundwater under agricultural soil	2.66	(µg/L)	

Table 6 Calculated risk characterisation ratios (RCR) for the local scenario.

Effected compartments and target organisms	Calculated risk characterisation ratios (RCR)	
the local fresh-water compartment	16,1	
the local fresh-water sediment compartment	0,09	
the local soil compartment	0,14	
the sewage treatment plant	9,98E-04	
fish-eating birds and mammals (fresh-water)	3,50E-03	
worm-eating birds and mammals (3 Mg/ha/year	8,63	
worm-eating birds and mammals (15 Mg/ha/year	43,2	
Notes.

* maximum sewage sludge application in agriculture and for land reclamation for agricultural purposes.

** maximum sewage sludge application for the reclamation of land for non-agricultural purposes and adapting land to specific needs resulting from waste management plans, spatial development plans or decisions on land development and land use, for growing plants intended for the production of compost, for growing plants not intended for consumption and for the production of fodder.

Such variability indicates the need for more research to understand the multifactorial effects DEHP concentrations in leachate.

Seasonal changes

The highest measured DEHP concentrations in each year occurred during the lowest average seasonal precipitation: 394.4 g/L in autumn of 2014 with 16 mm, 73.9 g/L in the summer of 2015 with 33 mm, and 31.1 g/L in the spring of 2016 with 30 mm of precipitation.

It should be noted that 2015 was a year with a small amount of precipitation and prolonged drought during the summer, which might have affected measured DEHP concentrations.

In WCS of DEHP concentrations, only a seasonal correlation (−0.62) between concentrations and mean precipitation was found. Other places where the seasonality is observed have found no seasonal correlation related to DEHP concentrations; which may suggest landfill design and operations can prevent the effect of the seasonal changes. In the study by Asakura, Matsuto & Tanaka (2004) no seasonal or annual dependence of DEHP concentrations in leachate from landfills in Japan was observed.

Risk assessment

The developed algorithm of risk analysis related to phthalate emission in leachate from municipal landfills (Fig. 4) in EUSES is a widely-available tool for forecasting exposure to environmental elements at particular stages and can also be adopted for other contaminants in leachate from municipal landfills.

In our study, all leachate was collected and transported to the local municipal STP, and therefore ERA was carried out for DEHP in the leachate at the STP.

Our results indicated no risk for microorganisms of sewage treatment plants, along the conclusions of the EU RAR (Pakalin et al., 2008).

Obtained unacceptable potential risk for the local fresh-water compartment and for worm-eating birds and mammals (for sewage sludge applications) indicate the need to take risk reduction measures. Also the EU RAR (Pakalin et al., 2008) concluded that there is a need for limiting the risk posed by DEHP emissions to aquatic and terrestrial ecosystems (for sites processing polymers with DEHP or sites producing printing inks, sealants and/or adhesives with DEHP) and that there is a need for further information and/or testing.

In a recent study by Kotowska, Kapelewska & Sawczuk (2020) risk quotient values were calculated for DEHP with respect to effluent wastewater. Their results indicate that DEHP poses a high environmental risk for all trophic levels (algae and cyanobacteria, invertebrates, and fish) in effluent wastewaters considered in the study. The study also concluded that further action is needed to reduce the contamination of water with phthalates from landfills and sewage treatment plants.

As DEHP is trapped on sludge, due to its high log Kow (Tran et al., 2015), high concentrations may be present in treated sludge destined for agricultural use (Marttinen et al., 2003b). The Danish Veterinary and Food Administration reports that DEHP is not degraded in STP but mainly distributed to the sludge (Ladefoged, Nielsen & Muller, 2004). DEHP accumulation in sludge may restrict its agricultural usage (Marttinen, Kettunen & Rintala, 2003a): the maximum acceptable value of 100 µg/g d.w. for sludge to be used in agriculture proposed by the European Commission (Marttinen et al., 2003b) is not incorporated into Polish sludge regulations. If sludge is uncontrolled (both in terms of the amount of applied doses and the preparation of the sludge prior to use), risks to the soil and groundwater from DEHP pollution remains high. In that context, the monitoring of DEHP in sludge and agricultural soil may contribute to improvements in the regulation (inter alia in Poland) recommending maximum levels of DEHP contaminants in sludge for agricultural application.

Conclusions

We estimated DEHP emissions from one MSW landfill over a 3-year period, investigated the seasonal changes of its concentrations in the leachate, and estimated, with the use of EUSES modeling, the associated potential risks to environmental components. DEHP was a ubiquitous environmental contaminant. Concentrations in leachates were in the range of < LOQ to 394.4 μg/L, depending on the sampling year and season. The highest measured DEHP concentrations in each year occurred during the lowest average seasonal precipitation.

The authors’ algorithm of risk analysis related to phthalate emission in leachate from MSW landfills is a widely-available tool for forecasting exposure to environmental elements at particular stages, which can be used by others interested in a risk analysis context.

Uncontrolled sludge management increases the risk to soil and groundwater posed by DEHP pollution. In this study the risk analyses was carried out for leachate from one MSW landfill in Pruszków, which directs all its leachates to an STP. The risk analysis indicated that for selected species of freshwater organisms and for birds and mammals feeding on earthworms (for sewage sludge applications), there are potentially unacceptable risks and a more detailed assessment should be considered.

With approximately one million Mg of DEHP expected to end up in Polish MSW landfills, DEHP pollution will pose a problem in the future.

More research on the potential risks of DEHP in leachate is needed and detailed modelling should be conducted to guide Polish policy in regards to landfill leachate, sewage sludge and wastewater.

Supplemental Information

Supplemental Information 1 Analytical methods and raw data for Fig. 1

Click here for additional data file.

Supplemental Information 2 ESUSES modeling report 1

Click here for additional data file.

Supplemental Information 3 ESUSES modeling report 2

Click here for additional data file.

Supplemental Information 4 Raw data for Table 3 and Fig. 2

Click here for additional data file.

The authors of this study would like to thank the late Andrzej Barański for his inspiration and guidance, Paulina Chaber for analytical support and Andres Araujo.

Additional Information and Declarations

Competing Interests

Author Contributions

Field Study Permissions

Data Availability

The authors declare there are no competing interests.

Paweł Wowkonowicz conceived and designed the experiments, performed the experiments, analyzed the data, prepared figures and/or tables, authored or reviewed drafts of the paper, and approved the final draft.

Marta Kijeńska and Eugeniusz Koda analyzed the data, authored or reviewed drafts of the paper, and approved the final draft.

The following information was supplied relating to field study approvals (i.e., approving body and any reference numbers):

President of MZO Pruszków (Municipal solid waste landfill Pruszków) has given consent for groundwater and leachate sampling.

The following information was supplied regarding data availability:

The raw data is available in the Supplemental Files.

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
