# Peer review of "Potential environmental risk assessment of di-2-ethylhexyl phthalate emissions from a municipal solid waste landfill leachate"

_PeerJ, doi:10.7717/peerj.12163_

## Round 0.1 · original submission · Major Revisions

My apologies for the length of time your manuscript was in review. The opinions of the reviewers are mixed, and I was never able to get a 3rd reviewer. Neither of the 2 reviews has much detail or many specific comments. There are concerns about the contribution of the research, which are amplified by further comments about repeated text and lengthy generic statements in the results, discussion, and conclusion sections. Please address these concerns and others pointed out in the reviews.

Reviewer 1 ·

Basic reporting

1. Sentences used in this work were too verbose.
2. A lot of repetitions were in the results and discussion, and these two parts should be combined and reorganized.

Experimental design

no comment

Validity of the findings

This paper lacks of novelty and a lot of similar works have been reported. DEHP in landfill leachate was detected in 3-years periods and PNEC was used to assess the eco-risk of leachate. Only PNEC was calculated, which was not supported. More toxicity test should be employed here.

Additional comments

Line 248-255: These sentences were too verbose and you should come straight to the results.
Line 250-251: The description of PNEC should be put into the method part.
Line 278-281: More explanation should be done here and why DEHP concentrations can significantly differ, even in the same country?
Line 291-295: This work compared DEHP concentrations in landfill leachate from different regions. The explanation is too confusing and inexplicable.

The conclusion is too long and lacks originality.

Reviewer 2 ·

Basic reporting

The topic is not specific on what element of risk assessment the author was focusing on. Please incorporate the element that the risk assessment is related to. This is for more clarity to the readers and not make the topic to generic.

Also the Topic should be “potential” risk assessment because actual risk assessment was not conducted to avoid misleading the readers

The introduction was well written with an extensive explanation of DEHP and its potential environmental and health risk if in contact. However, nothing was said on the national strategies and limits of the quantity of DEHP in leachate or ground water posed by the polish or European government.

Experimental design

Line 155, state the meteorological seasons in parenthesis for more clarity

Line 183-185, Why was the instrument changed and what was the difference the author identified while using the two instrument.

Validity of the findings

was written in a proper scientific manner

---

## Round 0.2 · accepted · Accept

Thank you for your efforts in revising the manuscript in response to reviewer comments.